# NATURE-INSPIRED LOCAL PROPAGATION

## ABSTRACT

The spectacular results achieved in machine learning, including the recent advances in generative AI, rely on large data collections. On the opposite, intelligent processes in nature arises without the need for such collections, but simply by on-line processing of the environmental information. In particular, natural learning processes rely on mechanisms where data representation and learning are intertwined in such a way to respect spatiotemporal locality. This paper shows that such a feature arises from a "pre-algorithmic" view of learning that is inspired by related studies in Theoretical Physics and adopts a description in terms of continuous time laws. We show that the algorithmic interpretation of the derived "laws of learning", which takes the structure of Hamiltonian equations, reduces to Backpropagation when the the speed of propagation goes to infinity. This opens the doors to machine learning studies based on full on-line information processing that are based the replacement of Backpropagation with the proposed spatiotemporal local algorithm.

## 1 INTRODUCTION

By and large, the spectacular results of Machine Learning in nearly any application domain are strongly relying on large data collections along with associated professional skills. Interestingly, the successful artificial schemes that we have been experimenting under this framework are far away from the solutions that Biology seems to have discovered. We have recently seen a remarkable effort in the scientific community of explore biologically inspired models (e.g. see Scellier & Bengio (2017), Kendall (2021); Scellier (2021); Laborieux & Zenke (2022)) where the crucial role of temporal information processing it is clearly identified.

While this paper is related to those investigations, it is based on more strict assumptions on environmental interactions that might stimulate efforts towards a more radical transformation of machine learning with emphasis on temporal processing. In particular, we assume that learning and inference develop jointly under a nature based protocol of environmental interactions and then we suggest developing computational learning schemes regardless of biological solutions. Basically, the agent is not given the privilege of recording the temporal stream, but only to represent it properly by appropriate abstraction mechanisms. While the agent can obviously use its internal memory for storing those representations, we assume that it cannot access data collection. The agent can only use buffers of limited dimensions for storing the processed information. From a cognitive point of view, those small buffers allows the agent to process the acquired information backward in time by implementing a sort of focus of attention. We propose a pre-algorithmic approach which derives from the formulation of learning as an Optimal Control problem and propose an approach to its solution that is also inspired by principles of Theoretical Physics. We formulate the continuous learning problem to emphasize how optimization theory brings out solutions based on differential equations that recall similar laws in nature. The discrete counterpart, which is more similar to recurrent neural network algorithms that can be found in the literature, can promptly be derived by numerical solutions. Interestingly, we show that the on-line computation described in the paper yields spatiotemporal locality, thus addressing also the longstanding debate on Backpropation biological plausibility (see Crick (1989); Lillicrap et al. (2016); Scellier & Bengio (2019)) by overcoming the unrealistic assumption of instantaneous propagation throughout the network. Finally, the paper shows that the conquest of locality opens up a fundamental problem, namely that of approximating the solution of Hamilton's equations with boundary conditions using only initial conditions. A few insights on the solution of this problem are given for the task of tracking in optimal control, which opens the doors of a massive investigation of the proposed approach.

## 2 RECURRENT NEURAL NETWORKS AND SPATIOTEMPORAL LOCALITY

We put ourselves in the genral case where the computational model that we are considering is based on a digraph $D = (V, A)$ where $V = \{1, 2, \ldots, n\}$ is the set of vertices and and $A$ is the set of directed arches that defines the structure of the graph. Let $\mathrm{ch}(i)$ denote the set of vertices that are childrens of vertex $i$ and with $\mathrm{pa}(i)$ the set of vertices that are parents of vertex $i$ for any given $i \in V$. More precisely we are interested in the computation of neuron outputs over a temporal horizon $[0, T]$. Formally this amounts to determine to assign to each vertex $i \in V$ a trajectory $x_i$ that is computed parametrically in terms of the other neuron outputs and in terms of an environmental information, mathematically represented by a trajectory[1] $u \colon [0, +\infty) \to \mathbb{R}^d$. We will assume that the output of the first $d$ neurons (i.e the value of $x_i$ for $i = 1, \ldots, d$) matches the value of the components of the input: $x_i(t) = u_i(t)$ for $i = 1, \ldots d$ and $\forall t \in [0, T]$. In order to consistently interpret the first $d$ neurons as input we require two additional property of the graph structure:

$$\mathrm{pa}(i) = \emptyset \quad \forall i = 1, \ldots, d; \tag{1}$$

$$\mathrm{pa}(\{d+1, \ldots, n\}) \supset \{1, \ldots, d\}. \tag{2}$$

Here equation 1 says that an input neuron do not have parents, and it also implies that no self loops are allowed for the input neurons. On the other hand equation 2 means that all input neurons are connected to at least one other neuron amongst $\{d+1, \ldots, n\}$.

We will denote with $x(t)$ (without a subscript) the ordered list of all the output of the neurons at time $t$ except for the input neurons, $x(t) := (x_{d+1}(t), \ldots, x_n(t))$, and with this definition we can represent $x(t)$ for any $t \in [0, T]$ as a vector in the euclidean space $\mathbb{R}^{n-d}$. This vector is usually called the *state* of the network since its knowledge gives you the precise value of each neuron in the net. The parameters of the model are instead associated to the arcs of the graph via the map $(j, i) \in A \mapsto w_{ij}$ where $w_{ij}$ assumes values on $\mathbb{R}$. We will denote with $w_{i*}(t) \in \mathbb{R}^{|\mathrm{pa}(i)|}$ the vector composed of all the weights corresponding to arches of the form $(j, i)$. If we let $N := \sum_{i=1}^{n} |\mathrm{pa}(i)|$ the total number of weights of the model we also define $\mathbb{R}^N \ni \boldsymbol{w}(t) := (w_{1*}(t), \ldots, w_{n*}(t))$ the concatenation of all the weights of the network. Finally we will assume that the output of the model is computed in terms of a subset of the neurons. More precisely we will assume that a vector of $m$ indices $(i_1, \ldots, i_m)$ with $i_k \in \{d+1, \ldots, n\}$ and at each temporal instant the output of the net is a function $\pi \colon \mathbb{R}^m \to \mathbb{R}^h$ of $(x_{i_1}, \ldots, x_{i_m})$, that is $\pi(x_{i_1}, \ldots, x_{i_m})$ is the output of our model. For future convenience we will denote $O = \{i_1, \ldots, i_m\}$.

**Temporal locality and causality** In general we are interested in computational schemes which are both local in time and causal. Let us assume that we are working at some fixed temporal resolution $\tau$, meaning that we can define a partition of the half line $(0, +\infty)$, $\mathcal{P} := \{0 = t_\tau^0 < t_\tau^1 < \cdots < t_\tau^n < \ldots\}$ with $t_\tau^n = t_\tau^{n-1} + \tau$, then the input signal becomes a sequence of vectors $(U_\tau^n)_{n=0}^{+\infty}$ with $U_\tau^n := u(t_\tau^n)$ and the neural outputs and parameters can be regarded as an approximation of the trajecotries $x$ and $\boldsymbol{w}$: $X_\tau^n \approx x(t_\tau^n)$ and $W_\tau^n \approx \boldsymbol{w}(t_\tau^n)$ $n = 1, \ldots \lfloor T/\tau \rfloor$. A local computational rule for the neural outputs means that $X_\tau^n$ is a function of $X_\tau^{n-l}, \ldots, X_\tau^n, \ldots, X_\tau^{n+l}, W_\tau^{n-l}, \ldots, W_\tau^n, \ldots, W_\tau^{n+l}$ and $t_\tau^{n-l}, \ldots, t_\tau^n, \ldots, t_\tau^{n+l}$, where $l \ll T/\tau$ can be thought as the order of locality. If we assume that $l \equiv 1$ (first order method) which means that

$$X_\tau^n = F(X_\tau^{n-1}, X_\tau^n, X_\tau^{n+1}, W_\tau^{n-1}, W_\tau^n, W_\tau^{n+1}, t_\tau^{n-1}, t_\tau^n, t_\tau^{n+1}). \tag{3}$$

Causality instead express the fact that only past information can influence the current state of the variables meaning that actually equation 3 should be replaced by $X_\tau^n = F(X_\tau^{n-1}, W_\tau^{n-1}, t_\tau^{n-1})$. Going back to the continuous description this equation can be interpreted as a discretization of

$$\dot{x} = f(x, \boldsymbol{w}, t), \tag{4}$$

with initial conditions.

---

[1]In the reminder of the paper we will try whenever possible to formally introduce functions by clearly stating domain a co-domain. In particular whenever the function acts on a product space we will try to use a consistent notation for the elements in the various sets that define the input so that we can re-use such notation to denote the partial derivative of such function. For instance let us suppose that $f \colon A \times B \to \mathbb{R}$ is a function that maps $(a, b) \mapsto f(a, b)$ for all $a \in A$ and $b \in B$. Then we will denote with $f_a$ *the function* that represents the partial derivative of $f$ with respect to its first argument, with $f_b$ the partial derivative of $f$ with respect to its second argument *as a function* and so on. We will instead denote, for instance, with $f_a(x, y)$ the element of $\mathbb{R}$ that represent the value of $f_a$ on the point $(x, y) \in A \times B$.

**Spatial locality** Furthermore we assume that such computational scheme is local in time and make use only on spatially local (with respect to the structure of the graph) quantities more precisely has the following structure

$$\begin{cases} x_i(t) = u_i(t) & \text{for } i = 1, \ldots d \quad \text{and} \quad \forall t \in [0, T]; \\ c_i^{-1}\dot{x}_i(t) = \Phi^i(x_i(t), \text{PA}^i(x(t)), \text{IN}^i(\boldsymbol{w}(t))) & \text{for } i = d+1, \ldots, n \quad \text{and} \quad \forall t \in [0, T], \end{cases} \quad (5)$$

Where $c_i > 0$ for all $i = d+1, \ldots, n$ set the velocity constant of that controls the updates of the $i$-th neuron, $\Phi^i \colon \mathbb{R} \times \mathbb{R}^{|\text{pa}(i)|} \times \mathbb{R}^{|\text{pa}(i)|} \to \mathbb{R}$ for all $i = d+1, \ldots, n$ performs the mapping $(r, \alpha, \beta) \mapsto \Phi^i(r, \alpha, \beta)$ for all $r \in \mathbb{R}$, $\alpha, \beta \in \mathbb{R}^{|\text{pa}(i)|}$, $\text{PA}^i \colon \mathbb{R}^{n-d} \to \mathbb{R}^{|\text{pa}(i)|}$ project the vector $\xi \in \mathbb{R}^{n-d} \mapsto \text{PA}^i(\xi)$ on the subspace generated by neurons which are in $\text{pa}(i)$ and $\text{IN}^i \colon \mathbb{R}^N \to \mathbb{R}^{|\text{pa}(i)|}$ maps the any vector $\boldsymbol{\omega} \in \mathbb{R}^N \mapsto \text{IN}^i(\boldsymbol{\omega})$ onto the space spanned by only the weights associated to arcs that points to neuron $i$. The assumptions summarized above describe the basic properties of a RNN or, as sometimes is referred to when dealing with a continuous time computation, a Continuous Time RNN Sompolinsky et al. (1988). The typical form of function $\Phi_i$, is the following

$$\Phi^i(r, \alpha, \beta) = -r + \sigma(\beta \cdot \alpha), \quad \forall r \in \mathbb{R} \quad \text{and} \quad \forall \alpha, \beta \in \mathbb{R}^{|\text{pa}(i)|}. \quad (6)$$

where in this case $\cdot$ is the standard scalar product on $\mathbb{R}^{|\text{pa}(i)|}$ and $\sigma \colon \mathbb{R} \to \mathbb{R}$ is a nonlinear bounded smooth function (usually a sigmoid-like activation function). Under this assumption the state equation in equation 5 becomes

$$c_i^{-1}\dot{x}_i(t) = -x_i(t) + \sigma(\text{IN}^i(\boldsymbol{w}(t)) \cdot \text{PA}^i(x(t))) \equiv -x_i(t) + \sigma\Big(\sum_{j \in \text{pa}(i)} w_{ij}x_j(t)\Big), \quad (7)$$

which is indeed the classical neural computation. Here we sketch a result on the Bounded Input Bounded Output (BIBO) stability of this class of recurrent neural network which is also important for the learning process that will be described later.

**Propostion 1.** *The recurrent neural network defined by ODE (7) is (BIBO) stable.*

*Proof.* See Appendix D ☐

## 3 LEARNING AS A VARIATIONAL PROBLEM

In the computational model described in Section 2, once the graph $D$ and an input $u$ is assigned, the dynamics of the model is determined solely by the functions that describes the changes of the weights over time. Inspired by the Cognitive Action Principle Betti et al. (2019) that formulate learning for FNN in terms of a variational problem, we claim that in an online setting the laws of learning for recurrent architectures can also be characterized by minimality of a class of functional. In what follows we will then consider variational problems for a functional of the form

$$F(\boldsymbol{w}) = \int_0^T \Big[\frac{mc}{2}|\dot{\boldsymbol{w}}|^2 + c\ell(\boldsymbol{w}(t), x(t; \boldsymbol{w}), t)\Big]\phi(t)\,dt, \quad (8)$$

where $x(\cdot, \boldsymbol{w})$ is the solution of equation 4 with fixed initial conditions[2], $\phi \colon [0, T] \to \mathbb{R}$ is a strictly positive smooth function that weights the integrand, $m > 0$, $\ell \colon \mathbb{R}^n \times \mathbb{R}^N \times [0, T] \to \overline{\mathbb{R}}_+$ is a positive function, that later on in this section will be interpreted as a loss function, and finally $c := \sum_{i=d+1}^n c_i/(n-d)$. We discuss under which conditions the stationarity conditions of this class of functional can be made temporally and spatially local and how they can be interpreted as learning rules.

### 3.1 OPTIMAL CONTROL APPROACH

The problem of minimizing the functional in equation 8 can be solved by making use of the formalism of Optimal Control. The first step is to put this problem in the canonical form by introducing an additional control variable as follow

$$G(\boldsymbol{v}) = \int_0^T \Big[\frac{mc}{2}|\boldsymbol{v}|^2 + c\ell(\boldsymbol{w}(t; \boldsymbol{v}), x(t; \boldsymbol{v}), t)\Big]\phi(t)\,dt, \quad (9)$$

---

[2]We do not explicitly indicate the dependence on the initial condition to avoid cumbersome notation.

where $\boldsymbol{w}(t; \boldsymbol{v})$ and $x(t; \boldsymbol{v})$ solve

$$\dot{x}(t) = f(x(t), \boldsymbol{w}(t), t), \quad \text{and} \quad \dot{\boldsymbol{w}}(t) = \boldsymbol{v}(t). \tag{10}$$

Then the minimality conditions can be expressed in terms of the Hamiltonian function (see Appendix A):

$$H(\xi, \boldsymbol{\omega}, p, q, t) = -\frac{1}{\phi(t)} \frac{q^2}{2mc} + c\ell(\boldsymbol{\omega}, \xi, t)\phi(t) + p \cdot f(\xi, \boldsymbol{\omega}, t), \tag{11}$$

via the following general result.

**Theorem 1** (Hamilton equations). *Let $H$ be as in equation 11 and assume that $x(0) = x^0$ and $\boldsymbol{w}(0) = \boldsymbol{w}^0$ are given. Then a minimum of the functional in equation 9 satisfies the Hamilton equations:*

$$\begin{cases} \dot{x}(t) = f(x(t), \boldsymbol{w}(t), t) \\ \dot{\boldsymbol{w}}(t) = -p_{\boldsymbol{w}}(t)/(mc\phi(t)) \\ \dot{p}_x(t) = -p_x(t) \cdot f_\xi(x(t), \boldsymbol{w}(t), t) - c\ell_\xi(\boldsymbol{w}(t), x(t), t)\phi(t) \\ \dot{p}_{\boldsymbol{w}}(t) = -p_x(t) \cdot f_{\boldsymbol{\omega}}(x(t), \boldsymbol{w}(t), t) - c\ell_{\boldsymbol{\omega}}(\boldsymbol{w}(t), x(t), t)\phi(t) \end{cases} \tag{12}$$

*together with the boundary conditions*

$$p_x(T) = p_{\boldsymbol{w}}(T) = 0. \tag{13}$$

*Proof.* See Appendix A     $\square$

## 3.2 RECOVERING SPATIO-TEMPORAL LOCALITY

Starting from the general expressions for the stationarity conditions expressed by equation 12 and equation 13, we will now discuss how the temporal and spatial locality assumptions that we made on our computational model in Section 2 leads to spatial and temporal locality of the update rules of the parameters $\boldsymbol{w}$.

**Temporal Locality** The local structure of equation 10, that comes from the locality of the computational model that we discussed in Section 2 guarantees the locality of Hamilton's equations 12. However the functional in equation 9 has a global nature (it is an integral over the whole temporal interval) and the differential term $m|\boldsymbol{v}|^2/2$ links the value of the parameters across near temporal instant giving rise to boundary conditions in equation 13. This also means that, strictly speaking equation 12 and equation 13 overall define a problem that is non-local in time. We will devote the entire Section 4 to discuss this central issue.

**Spatial Locality** The spatial locality of equation 12 directly comes from the specific form of the dynamical system in equation 5 and from a set of assumptions on the form of the term $\ell$. In particular we have the following result:

**Theorem 2.** *Let $\ell(\boldsymbol{\omega}, \xi, s) = kV(\boldsymbol{\omega}, s) + L(\xi, s)$ for every $(\boldsymbol{\omega}, \xi, s) \in \mathbb{R}^N \times \mathbb{R}^{n-d} \times [0, T]$, where $V: \mathbb{R}^N \times [0, T] \to \overline{\mathbb{R}}_+$ is a regularization term on the weights[3] and $L: \mathbb{R}^{n-d} \times [0, T] \to \overline{\mathbb{R}}_+$ depends only on the subset of neurons from which we assume the output of the model is computed, that is $L_{\xi_i}(\xi, s) = L_{\xi_i}(\xi, s)1_O(i)$, where $1_O$ is the indicator function of the set of the output neurons. Let $\Phi^i$ be as in equation 6 for all $i = d+1, \ldots, n$, then the generic Hamilton's equations described in equation 12 become*

$$\begin{cases} c_i^{-1}\dot{x}_i = -x_i + \sigma\Big(\sum_{j\in\mathrm{pa}(i)} w_{ij}x_j\Big) \\ \dot{w}_{ij} = -p_{\boldsymbol{w}}^{ij}/(mc\phi) \\ \dot{p}_x^i = c_i p_x^i - \sum_{k\in\mathrm{ch}(i)} c_k\sigma'\Big(\sum_{j\in\mathrm{pa}(k)} w_{kj}x_j\Big)p_x^k w_{ki} - cL_{\xi_i}(x, t)\phi \\ \dot{p}_{\boldsymbol{w}}^{ij}(t) = -c_i p_x^i \sigma'\Big(\sum_{m\in\mathrm{pa}(i)} w_{im}x_m\Big)x_j - ckV_{\boldsymbol{\omega}_{ij}}(\boldsymbol{w}, t)\phi \end{cases} \tag{14}$$

*Proof.* See Appendix B     $\square$

---

[3] a typical choice for this function could be $V(\boldsymbol{\omega}, s) = |\boldsymbol{\omega}|^2/2$ with $k > 0$

*Remark* 1. Notice equation 14 directly inherit the spatially local structure from the assumption in equation 5.

Theorem 2 other than giving us a manifestly spatio-temporal local structure shows that the computation of the $x$ costates has a very distinctive and familiar structure: for each neuron the values of $p_x^i$ are computed using quantities defined on chilren's nodes as it happens for the computations of the gradients in the Backpropagation algorithm for a FNN. In order to better understand the structure of equation 14 let us define an appropriately normalized costate

$$\lambda_x^i(t) := \frac{\sigma'(a_i(t))}{\phi(t)} p_x^i(t), \quad \text{with} \quad a_i(t) = \sum_{m \in \text{pa}(i)} w_{im} x_m \quad \forall i = d+1, \ldots, n, \qquad (15)$$

where we have introduced the notation $a_i(t)$ to stand for the activation of neuron $i$.[4] With these definitions we are ready to state the following result

**Propostion 2.** *The differential system in equation 14 is equivalent to the following system of ODE of mixed orders:*

$$\begin{cases} c_i^{-1} \dot{x}_i = -x_i + \sigma(a_i); \\ \ddot{w}_{ij} = -\frac{\dot{\phi}}{\phi} \dot{w}_{ij} + \frac{c_i}{mc} \lambda_x^i x_j + \frac{k}{m} V_{\boldsymbol{\omega}_{ij}}(\boldsymbol{w}, t); \\ \dot{\lambda}_x^i = \left[ -\frac{\dot{\phi}}{\phi} + \frac{d}{dt} \log(\sigma'(a_i)) + c_i \right] \lambda_x^i - \sigma'(a_i) \sum_{k \in \text{ch}(i)} c_k \lambda_x^k w_{ki} - c L_{\xi_i}(x, t) \sigma'(a_i), \end{cases} \qquad (16)$$

*where $\lambda_x^i$ is defined as in equation 15.*

*Proof.* See Appendix C ☐

This in an interesting result especially since via the following corollary gives a direct link between the rescaled costates $\lambda_x$ and the delta error of Backprop:

**Corollary 1** (Reduction to Backprop)**.** *Let $c_i$ be the same for all $i = 1, \ldots, n$ so that now $c_i = c$, then the formal limit of the $\dot{\lambda}_x$ equation in the system 16 as $c \to \infty$ is*

$$\lambda_x^i = \sigma'(a_i) \sum_{k \in \text{ch}(i)} \lambda_x^k w_{ki} + L_{\xi_i}(x, t) \sigma'(a_i). \qquad (17)$$

*Proof.* The proof comes immediately from equation 16. ☐

Notice that the equation for the $\lambda$ is exactly the update equation for delta errors in backpropagation: when $i$ is an output neuron its value is directly given by the gradient of the error, otherwise it is express as a sum on its childrens (see Gori et al. (2023)).

## 4   FROM BOUNDARY TO CAUCHY'S CONDITIONS

While discussing temporal locality in Section 3, we came across the problem of the left boundary conditions on the costate variables. We already noticed that these constraints spoil the locality of the differential equations that describe the minimality conditions of the variational problem at hand. In general, this prevents us from computing such solutions with a forward/causal scheme.

The following examples should suffice to explain that, in general, this is a crucial issue and should serve as motivation for the further investigation we propose in the present section.

**Example 1.** Consider a case in which $\ell(\boldsymbol{\omega}, \xi, s) \equiv V(\boldsymbol{\omega}, s)$, i.e. we want to study the minimization problem for $\int_0^T (m|\boldsymbol{v}(t)|^2/2 + V(\boldsymbol{w}(t; \boldsymbol{v}), t))\phi(t)dt$ under the constraint $\dot{\boldsymbol{w}} = \boldsymbol{v}$. Then the dynamical equation $\dot{x}(t) = f(x(t), \boldsymbol{w}(t))$ does not represent a constraint on variational problems

---

[4]We have avoided to introduce the notation till now because we believe that it is worth writing equation 14 with the explicit dependence on the variable $w$ and $x$ at least one to better appreciate its structure.

for functional in equation 9. If we look at the Hamilton equation for $\dot{p}_x$ in equation 12 this reduces to $\dot{p}_x = -p_x \cdot f_{\boldsymbol{\omega}}$. We would however expect $p_x(t) \equiv 0$ for all $t \in [0, T]$. Indeed this is the solution that we would find if we pair $\dot{p}_x = -p_x \cdot f_{\boldsymbol{\omega}}$ with its boundary condition $p_x(T) = 0$ in equation 13. Notice that however in general without this condition a random Cauchy initialization of this equation would not give null solution for the $x$ costate. Now assume that $\phi = \exp(\theta t)$ with $\theta > 0$, and $m = 1$. Assume, furthermore[5] that $V(\boldsymbol{\omega}, s) = |\boldsymbol{\omega}|^2/2$. The functional $\int_0^T (|\dot{w}|/2 + |w|^2/2)e^{\theta t}dt$ defined over the functional space[6] $H^1([0, T]; \mathbb{R}^N)$ is coercive and lower-semicontinuous, and hence admits a minimum (see Giaquinta & Hildebrandt (1995)). Furthermore one can prove (see Liero & Stefanelli (2013)) that such minimum is actually $C^\infty([0, T]; \mathbb{R}^N)$. This allows us to describe such minumum with the Hamilton equations described in equation 12. In particular as we already commented the relevant equations are only that for $\dot{w}$ and $\dot{p}_{\boldsymbol{w}}$ that is $\dot{w}(t) = -p_{\boldsymbol{w}}(t)e^{-\theta t}$ and $\dot{p}_{\boldsymbol{w}}(t) = -\boldsymbol{w}e^{\theta t}$ with $p_{\boldsymbol{w}}(T) = 0$. This first order system of equations is equivalent to the second order differential equation $\ddot{\boldsymbol{w}}(t) + \theta\dot{\boldsymbol{w}}(t) - \boldsymbol{w}(t) = 0$. Each component of this second order system will, in general have an unstable behaviour since one of the eigenvalues is always real and positive. This is a strong indication that when solving Hamilton's equations with an initial condition on $p_{\boldsymbol{w}}$ we will end up with a solution that is far from the minimum.

In the next subsection, we will analyze this issue in more detail and present some alternative ideas that can be used to leverage Hamilton's equations for finding causal online solutions.

## 4.1 TIME REVERSAL OF THE COSTATE

In Example 1 we discussed how the forward solution of Hamilton's equation 12 with initial conditions both on the state and on the costate in general cannot be related to any form of minimality of the cost function in equation 9 and this has to do with the fact that the proper minima are characterized also by left boundary conditions 13. The final conditions on $\dot{p}_x$ and $\dot{p}_{\boldsymbol{w}}$ suggest that the costate equations should be solved backward in time. Starting form the final temporal horizon and going backward in time is also the idea behind dynamic programming, which is of the main ideas at the very core of optimal control theory.

Autonomous systems of ODE with terminal boundary conditions can be solved "backwards" by time reversal operation $t \to -t$ and transforming terminal into initial conditions. More precisely the following classical result holds:

**Propostion 3.** *Let $\dot{y}(s) = \varphi(y(t))$ be a system of ODEs on $[0, T]$ with terminal conditions $y(T) = y^T$ and let $\rho$ be the time reversal transformation maps $t \mapsto s = T - t$, then $\hat{y}(s) := y(\rho^{-1}(s)) = y(t)$ satisfies $\dot{\hat{y}}(s) = -\varphi(\hat{y}(s))$ with initial condition $\hat{y}(0) = y^T$.*

Clearly equation 12 or equation 16 are not an autonomous system and hence we cannot apply directly Proposition 3 nonetheless, we can still consider the following modification of equation 14

$$\begin{cases} c_i^{-1}\dot{x}_i = -x_i + \sigma\Big(\sum_{j\in\mathrm{pa}(i)} w_{ij}x_j\Big) \\ \dot{w}_{ij} = -p_{\boldsymbol{w}}^{ij}/(mc\phi) \\ \dot{p}_x^i = -c_i p_x^i + \sum_{k\in\mathrm{ch}(i)} c_k\sigma'\Big(\sum_{j\in\mathrm{pa}(k)} w_{kj}x_j\Big)p_x^k w_{ki} + cL_{\xi_i}(x,t)\phi \\ \dot{p}_{\boldsymbol{w}}^{ij}(t) = c_i p_x^i \sigma'\Big(\sum_{m\in\mathrm{pa}(i)} w_{im}x_m\Big)x_j + ckV_{\boldsymbol{\omega}_{ij}}(\boldsymbol{w},t)\phi \end{cases} \tag{18}$$

which are obtained from equation 14 by changing the sign to $\dot{p}_x$ and $\dot{p}_{\boldsymbol{w}}$. Recalling the definition of the rescaled costates in equation 15 we can cast, in the same spirit of Proposition 2 a system of equations without $p_{\boldsymbol{w}}$. In particular we have as a corollary of Proposition 2 that

**Corollary 2.** *The ODE system in equation 18 is equivalent to*

$$\begin{cases} c_i^{-1}\dot{x}_i = -x_i + \sigma(a_i); \\ \ddot{w}_{ij} = -\dfrac{\dot{\phi}}{\phi}\dot{w}_{ij} - \dfrac{c_i}{mc}\lambda_x^i x_j - \dfrac{k}{m}V_{\boldsymbol{\omega}_{ij}}(\boldsymbol{w},t); \\ \dot{\lambda}_x^i = \Big[-\dfrac{\dot{\phi}}{\phi} + \dfrac{d}{dt}\log(\sigma'(a_i)) - c_i\Big]\lambda_x^i + \sigma'(a_i)\sum_{k\in\mathrm{ch}(i)} c_k\lambda_x^k w_{ki} + cL_{\xi_i}(x,t)\sigma'(a_i), \end{cases} \tag{19}$$

---

[5]The same argument that we give in this example works for a larger class of coercive potentials $V$.
[6]These are called Sobolev spaces, for more details see Brezis & Brézis (2011).

*Proof.* Let us consider equation 16. The change of sign of $\dot{p}_{\boldsymbol{w}}$ only affect the signs of $\lambda_x^i x_j$ and $V_{\boldsymbol{\omega}_{ij}}(\boldsymbol{w}, t)$ in the $\ddot{w}_{ij}$ equation, while the change of sign of $\dot{p}_x$ result in a sign change of the term $c_i \lambda_x^i$, $\sigma'(a_i) \sum_{k \in \text{ch}(i)} c_k \lambda_x^k w_{ki}$ and $L_{\xi_i}(x, t)\sigma'(a_i)$ in the equation for $\dot{\lambda}_i^x$. $\qquad\square$

Equation 19 is indeed particularly interesting because it offers an interpretation of the dynamics of the weights $w$ that is in the spirit of a gradient-base optimization method. In particular this allow us the extend the result that we gave in Corollary 1 to a full statement on the resulting optimization method

**Propostion 4** (GD with momentum). *Let $c_i$ be the same for all $i = 1, \ldots, n$ so that now $c_i = c$, and let $\phi(t) = \exp(\theta t)$ with $\theta > 0$ then the formal limit of the system in equation 19 as $c \to \infty$ is*

$$\begin{cases} x_i = \sigma(a_i); \\ \ddot{w}_{ij} = -\theta \dot{w}_{ij} - \frac{1}{m}\lambda_x^i x_j - (k/m)V_{\boldsymbol{\omega}_{ij}}(\boldsymbol{w}, t); \\ \lambda_x^i = \sigma'(a_i)\sum_{k \in \text{ch}(i)}\lambda_x^k w_{ki} + L_{\xi_i}(x, t)\sigma'(a_i). \end{cases} \tag{20}$$

*Remark* 2. This result shows that at least in the case of infinite speed of propagation of the signal across the network ($c \to \infty$) the dynamics of the weights prescribed by Hamilton's equation with the costate dynamics that is reversed (the sign of $\dot{p}_x$ and $\dot{p}_{\boldsymbol{w}}$ is changed) results in a gradient flow dynamic (heavy-ball dynamics) that it is interpretable as a gradient descent with momentum in the discrete. This is true since the term $\lambda_x^i x_j$ in this limit is exactly the Backprop factorization of the gradient of the term $L$ with respect to the weights.

In view of this remark we can therefore conjecture that also for $c$ fixed:

**Conjecture 1.** *Equation 19 is a* local optimization scheme *for the loss term $\ell$.*

Such result would enable us to use equation 19 with initial Cauchy conditions as desired.

## 4.2 CONTINUOUS TIME REVERSAL OF STATE AND COSTATE

Now we show that another possible approach to the problem of solving Hamilton's equation with Cauchy's conditions is to perform *simultaneous time-reversal* of both state and costate equation. Since in this case the sign flip involves both the Hamiltonian equations the approach is referred to as *Hamiltonian Sign Flip* (HSF). In order to introduce the idea let us begin with the following example.

**Example 2** (LQ control). Let us consider a linear quadratic scalar problem where the functional in equation 9 is $G(v) = \int_0^T qx^2/2 + rv^2/2\, dt$ and $\dot{x} = ax + bv$ with $q$, $r$ positive and $a$ and $b$ real parameters. The associated Hamilton's equations in this case are

$$\dot{x} = ax - sp, \quad \dot{p} = -qx - ap, \tag{21}$$

where $s \equiv -b^2/r$. These equation can be solved with the ansatz $p(t) = \theta(t)x(t)$, where $\theta$ is some unknown parameter. Differentiating this expression with respect to time we obtain

$$\dot{\theta} = (\dot{p} - \theta\dot{x})/x, \tag{22}$$

and using the equation 21 into this expression we find $\dot{\theta} - s\theta^2 - 2a\theta - q = 0$ which is known as *Riccati equation*, and since $p(T) = 0$, because of boundary equation 13 this implies $\theta(T) = 0$. Again if instead we try to solve this equation with initial condition we end up with an unstable solution. However $\theta$ solves an autonomous ODE with final condition, hence by Proposition 3 we can solve it with 0 initial conditions as long as we change the sign of $\dot{\theta}$. Indeed the equation $\dot{\theta} + s\theta^2 + 2a\theta + q = 0$ is asymptotically stable and returns the correct solution of the Riccati algebraic equation. Now the crucial observation is that, as we can see from equation 22, the sign flip of $\dot{\theta}$ is equivalent to the *simultaneous* sign flip of $\dot{x}$ and $\dot{p}$.

In Example 2, as we observe from equation 22, the sign flip of $\dot{\theta}$ is equivalent to the *simultaneous* sign flip of $\dot{x}$ and $\dot{p}$. Inspired by the fact, let us associate the general Hamilton's equation (equation 12), to this system the Cauchy problem

$$\begin{pmatrix} \dot{x}(t) \\ \dot{\boldsymbol{w}}(t) \\ \dot{p}_x(t) \\ \dot{p}_{\boldsymbol{w}}(t) \end{pmatrix} = s(t)\begin{pmatrix} f(x(t), \boldsymbol{w}(t), t) \\ -p_{\boldsymbol{w}}(t)/(mc\phi(t)) \\ -p_x(t) \cdot f_{\xi}(x(t), \boldsymbol{w}(t), t) - c\ell_{\xi}(\boldsymbol{w}(t), x(t), t)\phi(t) \\ -p_x(t) \cdot f_{\boldsymbol{\omega}}(x(t), \boldsymbol{w}(t), t) - c\ell_{\boldsymbol{\omega}}(\boldsymbol{w}(t), x(t), t)\phi(t) \end{pmatrix} \tag{23}$$

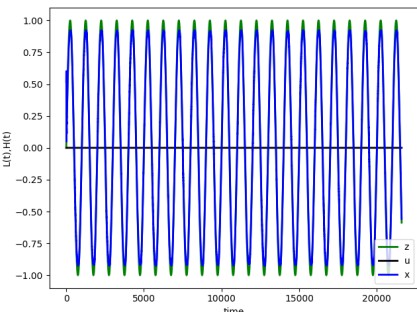
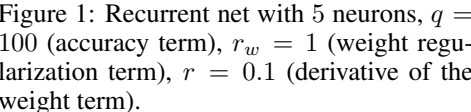

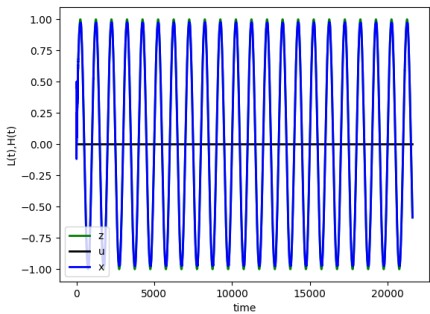

Figure 1: Recurrent net with 5 neurons, $q = 100$ (accuracy term), $r_w = 1$ (weight regularization term), $r = 0.1$ (derivative of the weight term).

Figure 2: Recurrent net with 5 neurons, $q = 1000$ (accuracy term), $r_w = 1$ (weight regularization term), $r = 0.1$ (derivative of the weight term)..

where for all $t \in [0, T]$, $s(t) \in \{0, 1\}$. Here we propose two different strategies that extends the sign flip discussed for the LQ problem.

**Hamiltonian Track** The basic idea is enforce system stabilization by choosing $s(t)$ to bound both the Hamiltonian variables. This leads to define an *Hamiltonian track*:

**Definition 1.** Let $S(\xi, \boldsymbol{\omega}, p, q) \subset (\mathbb{R}^{n-d} \times \mathbb{R}^N)^2$ for every $(\xi, \boldsymbol{\omega}, p, q) \in (\mathbb{R}^{n-d} \times \mathbb{R}^N)^2$ be a bounded connected set and let $t \mapsto X(t)$ any continuous trajectory in the space $(\mathbb{R}^{n-d} \times \mathbb{R}^N)^2$, then we refer to

$$\{(t, S(X(t)) : t \in [0, T]\} \in [0, T] \times (\mathbb{R}^{n-d} \times \mathbb{R}^N)^2$$

as *Hamiltonian track (HT)*.

Then we define $s(t)$ as follow

$$s(t) = \begin{cases} 1 & \text{if } (x(t), \boldsymbol{w}(t), p_x(t), p_{\boldsymbol{w}}(t)) \in S((x(t), \boldsymbol{w}(t), p_x(t), p_{\boldsymbol{w}}(t))) \\ -1 & \text{otherwise} \end{cases}. \tag{24}$$

For instance if we choose $S(\xi, \boldsymbol{\omega}, p, q) = \{(\xi, \boldsymbol{\omega}, p, q) : |\xi|^2 + |\boldsymbol{\omega}|^2 + |p|^2 + |q|^2 \leq R\}$ we are constraining the dynamics of equation 23 to be bounded since each time the trajectory $t \mapsto (x(t), \boldsymbol{w}(t), p_x(t), p_{\boldsymbol{w}}(t))$ moves outside of a ball of radius $R$ we are reversing the dynamics by enforcing stability.

**Hamiltonian Sign Flip Strategy and time reversal** We can easily see that the sign flip driven by the policy of enforcing the system dynamics into the HT corresponds with time reversal of the trajectory, which can nicely be interpreted as focus of attention mechanism. A simple approximation of the movement into the HT is that of selecting $s(t) = \text{sign}(\cos(\bar{\omega}t))$, where $\bar{\omega} = 2\pi\bar{f}$ is an appropriate *flipping frequency* which governs the movement into the HT. In the discrete setting of computation the strategy consists of flipping the right-side of Hamiltonian equations sign with a given period. In the extreme case the sign flip takes place at any Euler discretization step.

Here we report the application of the *Hamiltonian Sign Flip* strategy to the classic Linear Quadratic Tracking (LQT) problem by using a recurrent neural network based on a fully-connected digraph. The purpose of the reported experiments is to validate the HSF policy, which is in fact of crucial importance in order to exploit the power of the local propagation presented in the paper, since the proposed policy enables on-line processing.

The pre-algorithmic framework proposed in the paper, which is based on ODE can promptly give rise to algorithmic interpretations by numerical solutions. In the reported experiments we used Euler's discretization.
*Sinusoidal signals: The effect of the accuracy parameter.* In this experiment we used a sinusoidal target and a recurrent neural network with five neurons, while the objective function was $G(v) =$

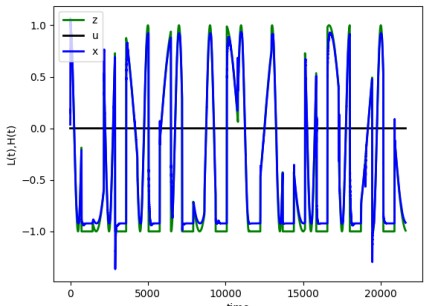

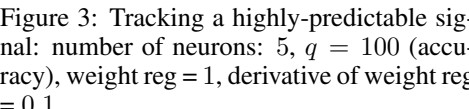

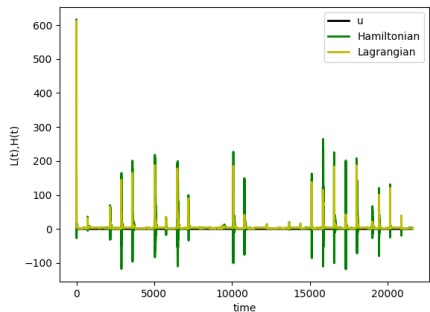

Figure 3: Tracking a highly-predictable signal: number of neurons: 5, $q = 100$ (accuracy), weight reg = 1, derivative of weight reg = 0.1

Figure 4: Evolution of the Lagrangian and of the Hamiltonian function for the experiment whose tracking is shown in the left-side figure.

$\int_0^T q(x - z)^2/2 + rv^2/2 + r_w^2 \, dt$, where we also introduced a regularization term on the weights. The HSF policy gives rise to the expected approximation results. In plot we can appreciate the effect of the increment of the accuracy term in Fig. 2.

*Tracking under hard predictability conditions* This experiment was conceived to assess the capabilities of the same small recurrent neural network with five neurons to track a signal which was purposely generated to be quite hard to predict. It is composed of patching intervals with cosine functions with constants. The massive experimental analysis on this and related examples confirms effectiveness of the HSF policy shown in Fig. 3. The side figure shows the behavior of the Lagrangian and of the Hamiltonian term. Interestingly, the last term gives us insights on the energy exchange with the environment.

## 5 CONCLUSIONS

This paper is motivated by the idea of a proposing learning scheme that, like in nature, arises without needing data collections, but simply by on-line processing of the environmental interactions. The paper gives two main contributions. First, it introduces a local spatiotemporal pre-algorithmic framework that is inspired to classic Hamiltonian equations. It is shown that the corresponding algorithmic formalization leads to interpret Backpropagation as a limit case of the proposed diffusion process in case of infinite velocity. This sheds light on the longstanding discussion on the biological plausibility of Backpropagation, since the proposed computational scheme is local in both space and time. This strong result is indissolubly intertwined with a strong limitation. The theory enables such a locality under the assumption that the associated ordinary differential equations are solved as a boundary problem. The second result of the paper is that of proposing a method for approximating the solution of the Hamiltonian problem with boundary conditions by using Cauchy's initial conditions. In particular we show that we can stabilize the learning process by appropriate schemes of time reversal that are related to focus of attention mechanisms. We provide experimental evidence of the effect of the proposed Hamiltonian Sign Flip policy for problems of tracking in automatic control.

While the proposed local propagation scheme is optimal in the temporal setting and overcomes the limitations of classic related learning algorithms like BPTT and RTRL, the given results show that there is no free lunch: The distinguishing feature of spatiotemporal locality needs to be sustained by appropriate movement policies into the Hamiltonian Track. We expect that other solutions better than the HSF policy herein proposed can be developed when dealing with real-word problems. This paper must only be regarded as a theoretical contribution which offers a new pre-algorithmic view of neural propagation. While the provided experiments support the theory, the application to real-world problems need to activate substantial joint research efforts on different application domains.

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

# A    OPTIMAL CONTROL

The classical way in which Hamilton's equations are derived is through Hamilton-Jacobi-Bellman theorem. So let enunciate this theorem in a general setting. Here we use the notation $y = (x, \boldsymbol{w})$ to stand for the whole state vector and $p = (p_x, p_{\boldsymbol{w}})$. We will also denote with $\alpha$ the control parameters. Moreover to avoid cumbersome notation in this appendix we will override the notation on the symbols $n$ and $N$ and we will use them here to denote the dimension of the state and of the control parameters respectively.

## A.1    HAMILTON JACOBI BELLMAN THEOREM

Consider the classical state model

$$\dot{y}(t) = f(y(t), \alpha(t), t), \quad t \in (t_0, T] \tag{25}$$

$f \colon \mathbb{R}^n \times \mathbb{R}^N \times [t_0, T] \to \mathbb{R}^n$ is a Lipschitz function, $t \mapsto \alpha(t)$ is the trajectory of the parameters of the model, which is assumed to be a *measurable function* with assigned initial state $y^0 \in \mathbb{R}^n$, that is

$$y(t_0) = y^0. \tag{26}$$

Let us now pose $\mathcal{A} := \{\alpha \colon [t_0, T] \to \mathbb{R}^N : \alpha \text{ is measurable}\}$ and given a $\beta \in \mathcal{A}$, and given an initial state $y^0$, we define the *state trajectory*, that we indicate with $t \mapsto x(t; \beta, y^0, t_0)$, the solution of equation 25 with initial condition equation 26.

Now let us define a cost functional $C$ that we want to minimize:

$$C_{y^0, t_0}(\alpha) := \int_{t_0}^{T} \Lambda(\alpha(t), y(t; \alpha, y^0, t_0), t) \, dt, \tag{27}$$

where $\Lambda(a, \cdot, s)$ is bounded and Lipshitz $\forall a \in \mathbb{R}^N$ and $\forall s \in [t_0, T]$. Then the problem

$$\min_{\alpha \in \mathcal{A}} C_{y^0, t_0}(\alpha) \tag{28}$$

is a constrained minimization problem which is usually denoted as *control problem* Bardi et al. (1997), assuming that a solution exists. The first step to address our constrained minimization problem is to define the *value function* or *cost to go*, that is a map $v \colon \mathbb{R}^n \times [t_0, T] \to \mathbb{R}$ defined as

$$v(\xi, s) := \inf_{\alpha \in \mathcal{A}} C_{\xi, s}(\alpha), \quad \forall (\xi, s) \in \mathbb{R}^n \times [t_0, T]$$

and the Hamiltonian function $H \colon \mathbb{R}^n \times \mathbb{R}^n \times [t_0, T] \to \mathbb{R}$ as

$$H(\xi, \rho, s) := \min_{a \in \mathbb{R}^N} \{\rho \cdot f(\xi, a, s) + \Lambda(a, \xi, s)\}, \tag{29}$$

being $\cdot$ the dot product. Then Hamilton-Jacobi-Bellman theorem states that

**Theorem 3** (Hamilton-Jacobi-Bellman). *Let us assume that $D$ denotes the gradient operator with respect to $\xi$. Furthermore, let us assume that $v \in C^1(\mathbb{R}^n \times [t_0, T], \mathbb{R})$ and that the minimum of $C_{\xi, s}$, Eq. equation 28, exists for every $\xi \in \mathbb{R}^n$ and for every $s \in [t_0, T]$. Then $v$ solves the PDE*

$$v_s(\xi, s) + H(\xi, Dv(\xi, s), s) = 0, \tag{30}$$

$(\xi, s) \in \mathbb{R}^n \times [t_0, T)$, *with terminal condition $v(\xi, T) = 0$, $\forall \xi \in \mathbb{R}^n$. Equation 30 is usually referred to as Hamilton-Jacobi-Bellman equation.*

*Proof.* Let $s \in [t_0, T)$ and $\xi \in \mathbb{R}^n$. Furthermore, instead of the optimal control let us use a constant control $\alpha_1(t) = a \in \mathbb{R}^N$ for times $t \in [s, s + \epsilon]$ and then the optimal control for the remaining temporal interval. More precisely let us pose

$$\alpha_2 \in \arg\min_{\alpha \in \mathcal{A}} C_{y(s + \varepsilon; a, \xi, s), s + \varepsilon}(\alpha).$$

Now consider the following control

$$\alpha_3(t) = \begin{cases} \alpha_1(t) & \text{if } t \in [s, s + \varepsilon) \\ \alpha_2(t) & \text{if } t \in [s + \varepsilon, T] \end{cases}. \tag{31}$$

Then the cost associated to this control is

$$
\begin{aligned}
C_{\xi,s}(\alpha_3) = &\int_s^{s+\varepsilon} \Lambda(a, y(t; a, \xi, s), t)\, dt \\
&+ \int_{s+\varepsilon}^T \Lambda(\alpha_2(t), y(t; \alpha_2, \xi, s), t)\, ds \\
= &\int_s^{s+\varepsilon} \Lambda(a, y(t; a, \xi, s), t)\, dt \\
&+ v(y(s+\varepsilon; a, \xi, s), s+\varepsilon)
\end{aligned}
\tag{32}
$$

By definition of value function we also have that $v(\xi, s) \leq C_{\xi,s}(\alpha_3)$. When rearranging this inequality, dividing by $\varepsilon$, and making use of the above relation we have

$$
\begin{aligned}
&\frac{v(y(s+\varepsilon; a, \xi, s), s+\varepsilon) - v(\xi, s)}{\varepsilon} + \\
&\frac{1}{\varepsilon} \int_s^{s+\varepsilon} \Lambda(a, y(t; a, \xi, s), t)\, dt \geq 0
\end{aligned}
\tag{33}
$$

Now taking the limit as $\varepsilon \to 0$ and making use of the fact that $y'(s, a, \xi, s) = f(\xi, a, s)$ we get

$$
v_s(\xi, s) + Dv(\xi, s) \cdot f(\xi, a, s) + \Lambda(a, \xi, s) \geq 0.
\tag{34}
$$

Since this inequality holds for any chosen $a \in \mathbb{R}^N$ we can say that

$$
\inf_{a \in \mathbb{R}^N} \{ v_s(\xi, s) + Dv(\xi, s) \cdot f(\xi, a, s) + \Lambda(a, \xi, s) \} \geq 0
\tag{35}
$$

Now we show that the $\inf$ is actually a $\min$ and, moreover, that minimum is 0. To do this we simply choose $\alpha^* \in \arg\min_{\alpha \in \mathcal{A}} C_{\xi,s}(\alpha)$ and denote $a^* := \alpha^*(s)$, then

$$
\begin{aligned}
v(\xi, s) = &\int_s^{s+\varepsilon} \Lambda(\alpha^*(t), y(t; \alpha^*, \xi, s), t)\, dt \\
&+ v(y(s+\varepsilon; \alpha^*, \xi, s).
\end{aligned}
\tag{36}
$$

Then again dividing by $\varepsilon$ and using that $y'(s; \alpha^*, \xi, a) = f(\xi, a^*, s)$ we finally get

$$
v_s(\xi, s) + Dv(\xi, s) \cdot f(\xi, a^*, s) + \Lambda(a^*, \xi, s) = 0
\tag{37}
$$

But since $a^* \in \mathbb{R}^N$ and we knew that $\inf_{a \in \mathbb{R}^N} \{ v_s(\xi, s) + Dv(\xi, s) \cdot f(\xi, a, s) + \Lambda(a, \xi, s) \} \geq 0$ it means that

$$
\begin{aligned}
&\inf_{a \in \mathbb{R}^N} \{ v_s(\xi, s) + Dv(\xi, s) \cdot f(\xi, a, s) + \Lambda(a, \xi, s) \} = \\
&\min_{a \in \mathbb{R}^N} \{ v_s(a, s) + Dv(\xi, s) \cdot f(\xi, a, s) + \Lambda(a, \xi, s \} = 0.
\end{aligned}
\tag{38}
$$

Recalling the definition of $H$ we immediately see that the last inequality is exactly (HJB). $\qquad \square$

### A.2 HAMILTON EQUATIONS: THE METHOD OF CHARACTERISTICS

Now let us define $p(t) = Dv(y(t), t)$ so that by definition of the value function $p(T) = 0$ which gives equation 13. Also by differentiating this expression with respect to time we have

$$
\dot{p}_k(t) = v_{\xi_k t}(y(t), t) + \sum_{i=1}^n v_{\xi_k \xi_i}(y(t), t) \cdot \dot{y}_i.
\tag{39}
$$

Now since $v$ solves equation 30, if we differentiate the Hamilton Jacobi equation by $\xi_k$ we obtain:

$$
v_{t\xi_k}(\xi, s) = -H_{\xi_k}(\xi, Dv(\xi, s), s) - \sum_{i=1}^n H_{\rho_i}(\xi, Dv(\xi, s), s) \cdot v_{\xi_k \xi_i}(\xi, s).
$$

Once we compute this expression on $(y(t), t)$ and we substitute it back into equation 39 we get:

$$
\dot{p}_k(t) = -H_{\xi_k}(y(t), Dv(y(t), t), t) + \sum_{i=1}^n \Big[ \dot{y}_i(t) - H_{\rho_i}(y(t), Dv(y(t), t), t) \Big] \cdot v_{\xi_k \xi_i}(y(t), t).
$$

Now if we choose $y$ so that it satisfies $\dot{y}(t) = H_\rho(y(t), p(t), t)$ the above equation reduces to

$$
\dot{p} = -H_\xi(y(t), p(t), t).
$$

Applying these equations to the Hamiltonian in equation 11 we indeed end up with equation 12.

## B  PROOF OF THEOREM 2

From equation 7 and the hypothesis on $\ell$ we have that

$$f^k_{\xi_i} = -c_i\delta_{ik} + c_k\sigma'\Big(\sum_{j\in\mathrm{pa}(k)} w_{kj}x_j\Big)\sum_{m\in\mathrm{pa}(k)} w_{km}\delta_{mi}, \qquad \ell_\xi = L_{\xi_i}(x,t)$$

$$f^k_{\boldsymbol{\omega}_{ij}} = c_k\sigma'\Big(\sum_{m\in\mathrm{pa}(k)} w_{km}x_m\Big)\sum_{h\in\mathrm{pa}(k)} \delta_{ik}\delta_{jh}x_h, \qquad \ell_{\boldsymbol{\omega}_{ij}} = kV_{\boldsymbol{\omega}_{ij}}.$$

Then equation 12 becomes

$$\begin{cases} c_i^{-1}\dot{x}_i = -x_i + \sigma\Big(\sum_{j\in\mathrm{pa}(i)} w_{ij}x_j\Big) \\ \dot{w}_{ij} = -p^{ij}_{\boldsymbol{w}}/(mc\phi) \\ \dot{p}^i_x = c_ip^i_x - \sum_{k=d+1}^n\sum_{m\in\mathrm{pa}(k)} c_kp^k_x\sigma'\Big(\sum_{j\in\mathrm{pa}(k)} w_{kj}x_j\Big)w_{km}\delta_{mi} - cL_{\xi_i}(x,t)\phi \\ \dot{p}^{ij}_{\boldsymbol{w}}(t) = -\sum_{k=d+1}^n c_kp^k_x\sigma'\Big(\sum_{m\in\mathrm{pa}(k)} w_{km}x_m\Big)\sum_{h\in\mathrm{pa}(k)} \delta_{ik}\delta_{jh}x_h - ckV_{\boldsymbol{\omega}_{ij}}(\boldsymbol{w},t)\phi \end{cases}$$

$$(40)$$

Now to conclude the proof it is sufficient to apply the following lemma to conveniently rewrite and switch the sums in the $\dot{p}$ equations.

**Lemma 1.** *Let $A$ be the set of the arches of a digraph as in Section 2, and let equation 2 be true, then*

$$A = \{\,(m,k)\in A : k\in\{d+1,\dots,n\}\,\} = \{\,(m,k)\in A : m\in\{1,\dots,n\}\,\}.$$

*Equivalently we may say that $\sum_{k=d+1}^n\sum_{m\in\mathrm{pa}(k)} = \sum_{m=1}^n\sum_{k\in\mathrm{ch}(m)}.$*

*Proof.* It is an immediate consequences of the fact that the first $d$ neurons are all parents of some neuron in $\{d+1,\dots,n\}$ (equation 2) and that they do not have themselves any parents (equation 1). $\qquad\square$

## C  PROOF OF PROPOSITION 2

The equations for $\ddot{w}_{ij}$ simply follows from differentiation of the expression for $\dot{w}_{ij}$ in equation 14 and the by the usage of the equation for $\dot{p}^{ij}_{\boldsymbol{w}}$ together with the definition for $\lambda^i_x$ in equation 15. The equation for $\dot{\lambda}$ in equation 16 instead can be obtained by differentiating with respect to time equation 15 and then using the expression of $\dot{p}^i_x$ from equation 14.

## D  PROOF OF PROPOSITION 1

Let $\mu(t) := \sigma\Big(\sum_{j\in\mathrm{pa}(i)} w_{ij}x_j(t)\Big)$ be. From the boundedness of $\sigma(\cdot)$ we know that there exists $B > 0$ such that $|\mu(t)| \le B$. Now we have

$$x_i(t) = x_i(0)e^{-\alpha t} + \int_0^t e^{-\alpha(t-\tau)}u(\tau)d\tau \le x_i(0) + B\int_0^t e^{-\alpha(t-\tau)}d\tau$$

$$\le x_i(0) + \frac{B}{\alpha}(1-e^{-t}) < x_i(0) + \frac{B}{\alpha}$$

