# OpenReview forum: "Nature-Inspired Local Propagation"
_ICLR.cc/2024/Conference — Submitted to ICLR 2024_

### Official Review · Reviewer_xnxp · 2023-10-27

**Soundness:** 2 fair
**Presentation:** 2 fair
**Contribution:** 2 fair
**Rating:** 5
**Confidence:** 3

**Summary:**

Summary:

The authors propose a framework that processes information online in a local spatiotemporal manner, motivated by biologically plausible learning schemes. They formulates continuous-time learning as an optimal control problem, connecting learning dynamics to Hamiltonian equations. They additionally show how taking the limit as propagation speed goes to infinity recovers backpropagation, suggesting backprop may emerge from local learning rules.  Ultimately, their framework requires solving Hamiltonian equations with boundary conditions, but the authors propose approximating the solution using only initial conditions via time reversal techniques.

**Strengths:**

Originality:

The authors present a novel perspective on learning by formulating it in the language of optimal control and Hamiltonian dynamics.  Using this language to connect backprop to local learning rules is, to my knowledge, novel.

Quality:

The paper seems theoretically grounded, with detailed mathematical derivations connecting backpropagation in simple neural networks to hamiltonian dynamics.

Clarity:

The paper is reasonably well structured and explains concepts like Hamiltonian equations and optimal control accessibly. I appreciated the numerous examples, and toy experiments.  The authors are likewise honest about the theoretical nature of the work.

Significance:

If validated, demonstrating backpropagation emerging from local learning would be quite significant.

**Weaknesses:**

It's very difficult to ascertain the significance of this work without further empirical validation.  I'd greatly encourage the authors to continue this line of work, and further to validate it on more mature learning problems.  I found the theoretical discussions to be somewhat tortuous, and the applicability of the method to be unclear.  The authors might try dramatically simplifying their presentation to make the thread between backpropagation and hamiltonian dynamics as clean as possible, and develop more experiments detailing the tradeoffs that the local learning rule implied by their work buys them.

**Questions:**

Could the authors describe how their local learning rule relates to BPTT in more detail?

Could the authors comment on the difficulty of applying their method to a more mature sequence learning problem, using, e.g., LSTMs over time-series data?

Can the authors comment on the time complexity of evaluating their method relative to backpropagation?

---

> ### Author Response · Authors · 2023-11-19
> **Response to Reviewer xnxp**
>
> The encouraging comment of  continuing this line of work is very
> much appreciated.
> We have been currently working on further validation
> ``on more mature learning problems'', but we believe that
> before presenting a professional application we need to prove
> that proposed theory is technically sound. Hence, this paper is
> primarily intended to provide the foundation of the theory. The
> experiments are primarily conceived to disclose the main principles more than
> for showing advances in application domains.
> We also fully understand the criticism on the presentation style.
> However, since this is the first paper which presents the theory,
> we decided not to run the risk of a rejection of the paper due to
> arguable statements. A comprehensible criticisms could have been
> that a new theory must not be claimed only by conjectures.
> In the near future, the Reviewer's comment will be carefully
> considered especially for reaching a broader scientific
> community.
>
> **Q1.** Our method reduces to Backprop and GD, as explained in the
> manuscript when we choose a directed acyclic graph and take the limit as $c
> \to \infty$ (classic feedforward nets used in deep learning).
> BPTT is not local in time and can only work for sequences of limited length,
> since we need to store neural activations and delta errors for all unfolded
> nets. The distinctive feaure of the neural propagation scheme that arises
> from Hamiltonian equations is that such a limitation is overcome!
>
> **Q2.** This is a crucial question that we have been considering of primary
> importance.
> We believe that the primary challenge in appropriately
> evaluating the proposed method lies in defining an experimental setting that
> does not distort the online/continual nature while still factoring out
> the effects introduced in this framework by forgetting phenomena.
> This because what we propose opens the doors to a way of conceiving
> experiments which can perfectly be understood when thinking of time-series
> data (exactly what the Reviewer was considering). Other applications in other
> domains of Machine Learning (e.g. vision, speech, and language) can be
> naturally considered.
>
> **Q3.** From a computational point of view we can find a strict
> relation between the computations required for BP/GD and our method. In
> particular with reference to Eq. (14) we have that:
>
> - The forward phase of BP is replaced by the residual-like computation
> of the state equation (the first equation of the system);
>
> - The backward phase that is responsible of the computation of the delta
> error in BP is here replaced by the computation of $\dot p_x$
> (third equation) that are
> the equivalent of the updates of the delta errors;
>
> - The computation of the gradients done with the BP factorization here
> is instead replaced by the computation of $\dot p_{\boldsymbol{w}}$ (forth
> equation);
>
> - Finally the gradient step is replaced by the update rule for the
> weights (second equation).
>
> Hence the time complexity of our proposal is the same as the BP algorithm.

---

### Official Review · Reviewer_yT4E · 2023-10-30

**Soundness:** 1 poor
**Presentation:** 1 poor
**Contribution:** 1 poor
**Rating:** 1
**Confidence:** 2

**Summary:**

I note that I am unable to provide a good summary of this paper, as I found it very difficult to understand. See below for more details on that.

The authors introduce a graph model of a computation, as a formalisation of a neural network. Although the model is continuous time, they also introduce a quantised time version of it. Their model has a notion of spatio-temporal locality, which allows them to express some constraints on the computation. They introduce a variational problem, although it wasn't clear to me what the relevance of this particular functional or its stationarity is. This allows the authors to derive a set of Hamiltonian equations. After this, I lost track of the direction the paper was going: the locality constraints were reconsidered, and there was a consideration of boundary conditions, followed by the introduction of some time-reversal transformations that I didn't understand in this context.

I wasn't able to understand how the conclusion related to the mathematics presented.

**Strengths:**

n/a

**Weaknesses:**

In my opinion the main weakness in this paper is that it does not explain itself adequately for the target audience. I'm not an expert in the field presented in the paper, but I am, I think, representative in terms of experience and knowledge of the ICLR audience, and I wasn't able to understand what this paper was trying to say. I think an ICLR paper should be able to: explain it's key finding, what this is, and why it's important to anyone in the audience; and should provide sufficient information for an interested non-specialist reader to understand the paper in detail if they are willing to put in the work. I don't feel that the paper met either of these criteria.

I would recommend that the authors either consider submitting the paper to a specialist journal, where the audience might be better matched to its content, or rewriting considerably to clearly explain the problem in the context of machine learning in general if they wish to submit to another general conference.

**Questions:**

My limited ability to understand the paper limits the number of questions I have.

One thing that stood out to me was the authors introduce the paper by noting that many ML approaches require a large amount of pre-collected data, whereas the techniques to be described in the paper will work "on-line", using a limited capacity memory to interpret the stream of information as it comes in in a causal way. This sounds a lot like reinforcement learning. Indeed there are some further resonances with reinforcement learning further through the paper where the authors mention dynamic programming, and their introduction of a Lagrangian-like variational objective, and the subsequent transformation into a Hamiltonian problem reminds one of the relationship between value functions and the local behaviour of optimal policies. I think if the authors wish to present to a broad audience it would be useful to address this question of the relationship to reinforcement learning, which is likely to come up.

---

> ### Author Response · Authors · 2023-11-19
> **Response to Reviewer yT4E**
>
> The Reviewer's comment on the presentation of the paper is understandable.
> We agree with the comment that the paper might not reach a broad
> audience. However, we are proposing a new approach that we believe to be relevant in the lifelong learning scenario.
> As such, we need to fully convince the scientific community of the technical
> soundness and, in particular, that the Hamiltonian equations yields a local
> spatiotemporal process which replaces Backpropation. While we considered to
> expose our results by adopting a more qualitative descriptions, we decided to
> give up so as not to run the risk of a rejection of the paper due to arguable
> statements. A comprehensible criticisms could have been that a new theory
> must not be claimed only by conjectures.  However, in the near future, our
> ambition is exactly what the Reviewer was asking: to present the paper to a
> broad audience and, especially to explore relationship to reinforcement
> learning.

---

### Official Review · Reviewer_gou7 · 2023-10-30

**Soundness:** 3 good
**Presentation:** 2 fair
**Contribution:** 3 good
**Rating:** 6
**Confidence:** 2

**Summary:**

The paper proposes a framework that treats an online learning setting for RNNs as a variational problem and allows to look at backpropagation as a special case. They use the Hamiltonian equations to represent the minimality conditions and propose an approximation method for the problem with boundary conditions.

**Disclaimer**: This paper is quite far from the area of my expertise (reinforcement learning, graph neural networks), and I won't be able to assess the technical excellence of the work,  I have not checked the proofs. However, I still think my review can be useful to the authors as an outsider's view, especially as the authors state in the conclusion that "the application to real-world problems need to activate substantial efforts on different application domains." Therefore, while not being totally ignorant in dynamical systems / optimal control, I ask the AC take my review as an evaluation of paper's readability and accessibility for the non-theoretical researchers that will potentially widen the target audience. I will also reflect this in the 'Confidence' section. For the time-being, I will put the score as 6, and, might update it post-rebuttal after engaging with the authors and other reviewers.

**Strengths:**

The framework proposed by the paper connects several seemingly unrelated concepts and allows to look at the commonly accepted things from a totally different angle (e.g. backpropagation).

**Weaknesses:**

* **Clarity**. While paper pays a lot of attention to careful notation and definition of the computational model, some of the concepts come out of the blue, either undefined or defined much later into the paper.
	* I understand that some of the concepts are not defined as they are common knowledge in the field (e.g. costates). However, my comment above is not only about those (see below)
	* What is pre-algorithmic view?
	* 'spatiotemporal locality' comes up already in the abstract/intro, but gets defined much later after all the computational model formalism is done.
	* I don't think, IN^i from (5) is defined.
* **Lack of examples**. I understand that it is common in the theoretical community to define things in a most general way possible. However, I think it will be highly beneficial to bring parallels to the machine learning case by giving examples even for the variables involved. For example, $l(w, x, t)$ comes up in equation 8, but only much later (Theorem 2 at the bottom of next page) the paper mentions that this can be a sum of the regularisation term and an indicator function on the neurons outputs multiplied by some function L, i.e this is a loss function! Same applies to other variables, e.g. $\xi$ and $\mathbf{u}$. Section 2 defines $u$ as d-dimensional and $\mathbf{w}$ as a concatenation of all weights in the network, i.e. n-dimensional. However, the function $l$ defined in (9) takes $\mathbf{w}, x, t$ as inputs, but function $l$ in (11) takes $\mathbf{u}, \xi, t$ as inputs. If I understand it correctly, these $\mathbf{u}$ is a control variable and is not the same as the input trajectory defined in Section 2. But having a clear example would greatly decrease the confusion.
* **Existing literature**. The paper is motivated by biological learning where the organisms do not have access to the entire data collection ("the agent can only use buffers of limited dimensions for storing the processed information") However, the paper ignores the existing literature on Online Learning and Online Optimisation, that are highly related to the topic. Putting the work into this context would be highly beneficial for the community.

**Questions:**

* Could you elaborate on the meaning of 'temporal horizon' in Section 2? Imagine we have an RNN which we update using gradient descent. The inputs have temporal aspect (e.g. sequences of real numbers), and the updates have temporal aspect (each weight updates adds one to the time subscript of the weight). Which one do you imply in Section 2? Do you intertwine the two as you mention in the second paragraph of Section 1 ("we assume that learning and inference develop jointly")? Could you provide a toy example?
* "we show that the on-line computation described in the paper yields spatiotemporal locality, thus addressing also the longstanding debate on Backpropagation biological plausibility". Could you provide a couple of references on the debate and add a sentence or two explaining how your result affect the debate.
* Section 2 defines the computational model based on a digraph. Should there also be an acyclicity constraint that does not allow "inputs -> A-> B->A" within the same time step?
* For me, equation 8 comes out of the blue. Section (2) introduces the computational model and the constraints on it. Beginning of Section 3 mention CAL (Betti et al., 2019) for FNN (feedforward NN?). And then you say "we claim that in an online setting the laws of learning for recurrent architectures can also be characterized by minimality of a class of functional. In what follows we will then concider variational problems for the functional of the form". Could you explain where does equation 8 come from? Is the mc/2 * |w|^2 the reqularisation term? If yes, does the $l$ in equation 8 differ from the $l$ in Theorem 2 that already includes the regularisation term?
* "The local structure of equation 10, that comes from the locality of the computational model from Section 2 guarantees the locality of Hamilton's equation 12". Could you, please, explain why this is the case?
* Beginning of page 5 mentions backprop for FNN leading to equation 15 after introducing normalised costates. Could you explain the jump from considering RNNs to FNNs?
* (Corollary 1) "the proof comes immediately from equation 16." Could you add a couple of explicit steps for me to understand why this is the case?
* "This is a strong indication that when solving Hamilton's equations with an initial condition on $p_w$ we will end up with a solution that is far from the minimum". How does this relate to any of the practical consequences on learning algorithms, e.g. backpropagation?
### Nits

* "Spectacular results achieved in machine learning, ..., rely on large data collections. On the opposite, intelligent processes in nature arise without the need for such collections"
	* True, but having access to libraries and archives have significantly boosted our ability to do science and fuelled the progress.
* page 5, "Theorem 2 other than giving us a spatio-temporal ... show". Is there a word missing here?
* footnote 5: "... class of cohercive potentials V" Should be "coercive"?

---

> ### Author Response · Authors · 2023-11-19
> **Response to Reviewer gou7 (part 1)**
>
> Many thanks to the Reviewer for his humble approach.  When looking at the
> constructive his criticisms, it looks like his/her expertise in
> reinforncement learning led to understanding the paper much more than what
> mentioned in the "Disclaimer part."
> We have tried to use some of the comments exposed in the weakness section
> to improve the presentation. In particular we tried to clarify some terms
> like `"pre-algorithmic algorithm" and to immediately comment on the role
> of loss function of the term $\ell$ right after Eq. (8). We furthermore
> changed the notation for the general variable that indicate the $\boldsymbol{w}$
> component of the state that was previously denoted as $\boldsymbol{u}$ and
> now it is named $\boldsymbol{\omega}$ to further disambiguate it from
> the input variable $u$. We would also like to remark that the ${\rm IN}$ function is
> defined on right after Eq. (5). In what follows we will answer to the precise
> questions that the Reviewer formulated in the corresponding section of
> his/her review.
>
> **Q1.** Yes, in this work, we are mainly concerned with online
> computations where there is a strong connection between the time that
> captures the dynamics of the sequence we are processing, expressed in our
> notation by the function $t \mapsto u(t)$, and the dynamics of the parameters
> of the model defined by the map $t \mapsto \boldsymbol{w}(t)$. Many learning
> protocols can be cast in this framework, ranging from reinforcement learning,
> lifelong and online continual learning.  The specific strong emphasis in this
> paper is that we look for a learning framework which can work without
> accessing large data collections, but simply by processing and properly
> representing the information as it is comes from sensors over the given
> temporal horizon (agent life span).
>
> **Q2.** We have added the references and commented how our
> approach contributes to the mentioned problem.
>
> **Q3.** Perhaps we understood the comment, but we invite the Reviewer
> to ask again if we didn't grasp the essence of the question.
> Cycles like $A\to B\to A$ amongst non-input units are permitted
> since our assumptions correspond to typical recurrent nets. However, the
> consistent computational scheme that we adopt does require that the
> processing does not take place "within the same time step"! In general, in
> the discrete setting of computation, we need a delay, while in the continuous
> framework this correspond to the introduction of temporal derivatives of the
> state.
>
> **Q4.** The term $\int_0^T
> \ell(\boldsymbol{w}(t),
> x(t;\boldsymbol{w}),t))\phi(t)\, dt,$
> comes from the ergodic transposition of the
> functional risk inspired by classical mechanics as discussed in
> (Betti et al., 2019). The basic idea is that when dealing with time and with a
> stream of information the expectation of the loss that defines the empirical
> risk can be rewritten as a sum over time. In our case however, differently
> from previous works in this directions, we are considering recurrent
> computations and, therefore, the term $x(t;\boldsymbol{w})$  is a solution of the
> recurrent dynamics expressed by Eq. (4). As correctly noticed by the Reviewer,
> function $\ell$ can itself contain a regularization term
> on the weights (like the classical weight decay). However it is important to
> add an additional *temporal regularization term* that links the weights
> at different temporal instant of the form $\int_0^T
> mc/2|\dot\boldsymbol{w}|^2\, \phi(t) dt$ for at least two reasons:
> firstly, it promotes
> convergence and the smoothness of the solution and, secondly, it enables
> the explicit calculation of the Hamiltonian in as in Eq. (11);
> hence, there is always a classic regularization term on the weights and also
> an appropriate temporal regularization term on the derivative of the weights
> that is consistently considered in Eq. (8) and in Theorem 2.
>
> **Q5.**
> This is because the fact that we are considering the particular set
> of differential constraints in Eq. (10) allows us (by using the
> techniques of optimal control) to describe the stationary point of
> (9) in terms again of local differential equations (Hamilton Equation).
> In this equation, in addition to the spatial locality, the presence of time
> derivatives gives rise to local temporal propagation.

---

> ### Author Response · Authors · 2023-11-19
> **Response to Reviewer gou7 (part 2)**
>
> **Q6.**
> While the reduction for $c\to\infty$ can be formally done in general
> as suggested for instance in Corollary 1, it only describes a meaningful and
> viable computational scheme only when we have a directed acyclic graph
> (which correspond to the FNN case).
> So the reduction indeed makes sense only under the additional hypothesis
> that are the one that makes possible to apply the standard Backpropagation
> algorithm.
>
> **Q7.** Consider the last equation in the system of Eq. (16) with all
> $c_i=c$, and group them as follows
> $$ \lambda^i_x -\sigma'(a_i)\sum_{k\in{\rm ch}(i)}\lambda^k_x w_{ki} - L_{\xi_i}(x,t)\sigma'(a_i)= \frac{1}{c} \dot \lambda^i_x -\frac{1}{c}\biggl[-\frac{\dot\phi}{\phi}+\frac{d}{dt}\log(\sigma'(a_i))\biggr] \lambda^i_x,$$
> now the rhs of this equation as $c\to\infty$ formally goes to $0$ and
> we are left with the desired equation.
>
> **Q8.** This implies that, in general, it is not possible to simply define
> learning/optimization theories that operate in a temporal environment solely
> in terms of optimization problems for integral functionals (like the one in
> Eq. 8), unless the problem is small enough to be solved in a batch-mode
> fashion. This observation is also related to the fact that the solution of
> calculus of variations typically describes elliptic problems, and they are
> incompatible with hyperbolic problems, which is usually the a class of
> problems that are suitable to describe temporal evolution (like learning in
> this context).
> This is why the discussion proposed in Section 4, addressing boundary
> conditions, is instrumental to enable the use of such tools within the
> context of learning.
>
> **"Nits"** We have address them directly in the revised manuscript.

---

### Official Review · Reviewer_YNF2 · 2023-11-01

**Soundness:** 3 good
**Presentation:** 2 fair
**Contribution:** 3 good
**Rating:** 5
**Confidence:** 1

**Summary:**

The paper develops a spatiotemporal local learning rule. To develop it authors treat a learning rule as a solution to a variational problem for the special class of functional that can be solved using Hamilton's equations. The authors show how this rule can be reduced to backpropagation and to backpropagation with momentum as specific cases.

**Strengths:**

I can not describe strengths and weaknesses based on my current understanding of the paper. I hope I will update these sections when the authors answer my questions.

Please see the questions section.

**Weaknesses:**

--

**Questions:**

1) Can you elaborate more how Corollary 1 can be viewed as backpropagation and Proposition 4 as GD with momentum?
2) Can you recommend an open source to understand how Corollary 1 can be viewed as backpropagation along with the Gori et al. (2023)?
3) Can a different choice of \phi lead to different learning rules in Proposition 4? Or does it lead only to different schedules for momentum?
4) Can you write algorithms for the proposed learning rules (including sign flipping algorithm)?
5) Can you motivate the form of functional in Eq. 8? Specifically, why regularization on weight acceleration is important?
6) How q x^2 /2 can be interpreted as the accuracy term in Section 4.2?
7) What are z, u and x in Fig. 1-4? z is target, x is a vector predicted with LSTM, and u is input? I failed to see how objectives in section 4.2 encourage x to be close to z. Can you elaborate on this?
8) You mentioned that the proposed algorithm overcomes the limitation of backpropagation through time, but at the same time the algorithm converges to backpropagation when c → to infinity. Can you elaborate on this?

**Details Of Ethics Concerns:**

--

---

> ### Author Response · Authors · 2023-11-19
> **Response to Reviewer YNF2 (part 1)**
>
> We thank the reviewer for his fairness in the evaluation and for having posed
> many specific question that will surely help to improve the quality of the
> manuscript.
>
> **Q1.** Corollary 1 shows that in the limit of infinite speed of
> propagation ($c\to+\infty$) the differential equation for the
> rescaled costates, the $\lambda_x^i$, described in Eq. (16) (last equation)
> satisfy exactly the classical equation for the delta error terms used in
> the Backpropagation algorithm. This reduction, which in Corollary 1
> is formally expressed in the general setting, make sense only in the
> case when the graph that we consider is acyclic which is an assumption
> required by Backprop. In this case, which correspond to the classical
> architectures of FNN
> we can choose, as it is usually done, the set of output nodes to be the
> one at the "end" or at the "top" of the architecture, i.e. the ones
> that have no childrens ($i\in O$ implies ${\rm ch}(i)=\emptyset$.
> Then if we look at Eq. (17):
> $$
> \lambda^i_x= \sigma'(a_i)\sum_{k\in{\rm ch}(i)}\lambda^k_x w_{ki} + L_{\xi_i}(x,t)\sigma'(a_i),
> $$
> we can make the following observations: *i.* if $i$ is an output neurons,
> $i\in O$, then the sum $\sum_{k\in{\rm ch}(i)}$ is empty and we get
> $$\lambda^i_x = L_{\xi_i}(x,t)\sigma'(a_i)$$
> which is exactly the statement that
> the delta error on the output neurons is simply the gradient of the loss
> with respect to their activation, *ii.* when $i\ne O$ it is
> the second term that vanishes since, as we remarked in Theorem 2 $L$
> only depends on the value of the neurons in the output and reduces to
> $$ \lambda^i_x= \sigma'(a_i)\sum_{k\in{\rm ch}(i)}\lambda^k_x w_{ki},$$
> that again is exactly the classical way in which the delta error is computed
> in Backprop.
>
> For what concerns Proposition 4 connections with GD with momentum can be
> better understood as follows. The second formula in Eq. (20) has the form
> $$\ddot w_{ij}+\theta \dot w_{ij} + \gamma \nabla_{u_{ij}} E(w,t)=0$$
> where $E$ is a generic compound loss.
> This is the case
> since the term $V_{u_{ij}}(\boldsymbol{w},t)$ is an explicit gradient of
> a regularization loss with respect to the weight $w_{ij}$ and $\lambda^i_x x_j$
> is the Backprop factorization  of the gradient of the loss
> that involves the outputs of the NN with respect again to the weight
> $w_{ij}$. If you consider an Euler discretization of this ODE
> you end up with
> $$\frac{1}{\tau}(w_{ij}^{k+1}-2 w_{ij}^k +w_{ij}^{k-1})
> +\theta(w_{ij}^{k+1}-w_{ij}^k)+\gamma \nabla_{u_{ij}} E(w^k, t_k)=0$$
> that can be rearranged into the GD with momentum update rule
> $$w_{ij}^{k+1}=w_{ij}^k -\alpha \nabla_{u_{ij}} E(w^k, t_k) +\beta(w^k_{ij}-w^{k-1}_{ij}).$$
> Interestingly this continuous interpretation
> was already introduced in the vary same paper that proposed for the
> first time the GD with momentum method, namely in
> B. T. Polyak, *U.S.S.R. Comput. Math. Math. Phys.* 4,1–17 (1964).
>
> **Q2.**
> As explained in the response to Q1, any textbook that explains in details the
> Backprop algorithm is ok. An excellent, widely available reference is the
> classical Rumelhart, David E., James L. McClelland, and PDP Research
> Group. "Parallel distributed processing, volume 1: Explorations in the
> microstructure of cognition: Foundations." (1986).
> For a completely open source material you can look for instance at this
> lecture note (section 6) available online
> https://www.cs.cornell.edu/courses/cs5740/2016sp/resources/backprop.pdf
> which appears to be ok but of which we cannot guarantee
> correctness of all its parts.
>
> **Q3.** The structure of the learning rule (not the precise form
> however!) that one can achieve with a general $\phi$ is described by
> Eq.~(16). A meaningful generalization of Proposition 4 with general $\phi$
> can be achieved if we choose $\phi$ such that $\dot\phi/\phi>0$. The effect
> on the resulting method is not simply that of having a time dependent
> coefficient in front of the momentum term as the choice of $\phi$ also
> influences the computation of the delta-term-like variables
> $\dot\lambda$. Hence, to some extent, the Reviewer's intuition that a truly
> different learning rule might arise makes sense. However, the concrete impact
> in the process of learning has still to be explored.

---

> ### Author Response · Authors · 2023-11-19
> **Response to Reviewer YNF2 (part 2)**
>
> **Q4.** The algorithmic description of the proposed rules
> basically relies on the explicit Euler approximation scheme of an ODE.
> [As a reference you can look at the classical book
> Richard, L., Faires J Douglas, and M. Annette.
> Numerical analysis. sengage, 2016.]
> For instance in the case of the ``sign-flipping algorithm'' the
> once you choose a temporal discretization step $\tau>0$ the algorithm looks
> like this
>
> 1. Randomly initialize the state and costate variables and set the sign
> term in Eq. (23) equal to $1$: $s\gets 1$;
> 2. Compute the update terms following Eq.(14):
>
>     - $\Delta x_i\gets c_i\Bigl(-x_i +\sigma\Bigl(\sum_{j\in{\rm pa}(i)} w_{ij}x_j\Bigr)\Bigr)$
>
>     - $\Delta w_{ij}\gets -p^{ij}_\boldsymbol{w}/(mc)$
>
>     - $\Delta p^i_x\gets c_i p_x^i-\sum_{k\in{\rm ch}(i)} c_k  \sigma'\Bigl(\sum_{j\in{\rm pa}(k)} w_{kj}x_j\Bigr)p_x^k w_{ki} - c L_{\xi_i}(x,t)$
>
>     - $\Delta p_{\boldsymbol{w}}^{ij}(t)\gets - c_i p^i_x \sigma'\Bigl(\sum_{m\in{\rm pa}(i)} w_{im}x_m\Bigr)x_j - c k V_{u_{ij}}(\boldsymbol{w},t)$
>
> 3. Update all the variables according to the implicit Euler scheme:
>
>     - $x_i\gets x_i +\tau s \Delta x_i$
>
>     - $w_{ij}\gets w_{ij} +\tau s \Delta w_{ij}$
>
>     - $p^i_x \gets p^i_x +\tau s \Delta p^i_x$
>
>     - $ p_{\boldsymbol{w}}^{ij} \gets p_\boldsymbol{w}^{ij}+\tau s \Delta p^{ij}_\boldsymbol{w}$$
>
> 4. Recompute  $s$ according to Eq. (24).
> 5.  If this is not the last iteration go back to step 2.
>
> **Q5.** The regularization on the velocity of the weights it is
> important for at least two reasons: firstly it promotes convergence in the
> parameters of the models, as well as the smoothness of the overall learning
> process, and secondly enables the explicit calculation of the Hamiltonian in
> as in Eq. (11); without this term the Hamiltonian would have been defined in
> terms of another implicit minimization problem.
>
> **Q6.** In the classical LQ problem the objective is to have bring the state variable
> as close as zero as possible, while also keeping small values of the control
> parameters, hence the more the term $x^2/2$ is weighted (by taking higher value
> of $q$) the more the solution is expected to accurately favour the zero,
> constant solution. In the experiment done with the sinusoidal signal there
> is indeed a misprint in the text the functional that we are considering is
> $G(v)=\int_0^T q (x-z)^2/2+rv^2/2 + r_w^2\, dt$ so that $q (x-z)^2/2$
> has a direct interpretation as accuracy parameter. This has been now
> corrected in the revised version.
>
> **Q7.** In this figure $x$ is the value of the output neuron,
> $z$ is the target and $u$ is an input as in Eq. (5) (notice that however in
> this experiment it is just a null signal). As explained in
> Q6, the actual accuracy term that is used in the experiments is $q (x-z)^2/2$
> which should clarify the Reviewer's legitimate question.
>
> **Q8.** The proposed theory works in general for a directed graph
> with the properties described in Section 2, which basically correspond with
> recurrent neural network architectures. However as we remarked also in Q1.,
> while the reduction for $c\to\infty$ can be formally done in general as
> suggested for instance in Corollary 1, it only describes a meaningful and
> viable computational scheme only when we have a directed acyclic graph (case
> for which Backprop can be used).
> So the reduction indeed makes sense only under the additional hypothesis at
> the basis of standard Backpropagation algorithm. In all other general cases,
> that are the one of interest for this study, the expression we found in the
> limit are just formal expressions. However, what is really important is that
> the proposed theory clearly leads to a spatiotemporal local propagation
> scheme which, to the best of our knowledge, is an open problem for recurrent
> neural nets.

---

### Author Response · Authors · 2023-11-19

First, we are pleased to inform the reviewers that, irrespective of this paper, their comments are very useful for our research activity. We greatly appreciate some constructive criticisms that reveal curiosity and an understanding of the essence of the paper, which might be higher than what is claimed in the ’confidence score’.

While the paper has been written with the long-term goal of contributing to the problem of learning from sequences without needing to store the processed information in data collections, the current experiments can only confirm that the proposed theory is technically sound. We acknowledge the importance of initiating the next steps for professional experimental analysis on machine learning benchmarks, but this requires additional efforts and is an objective of future research. As noted by some of the reviewers, there are also intriguing connections with reinforcement learning that are important subjects for further investigation.

The paper is primarily intended to disclose fundamental principles of neural propagation and internal representation more than facing application domains, which we are planned for a second step. The presentation style is affected by the need to address this issue formally by means of a theory which is technically sound. Regardless of the final decision on this acceptance at ICLR2024, the encouraging reviewers’ comments of continuing this line of work is very much appreciated.

Whenever possible, we have tried to incorporate reviewers’ suggestions directly into the revised version of the manuscript; such edits have also been commented on in the answers to the reviewers and highlighted in blue in the paper.

---

### Meta-Review · Area_Chair_8psW · 2023-12-08

**Metareview:**

The paper puts forward a theoretical foundation for time-continuous on-line information processing in recurrent neural-nets, which unlinke backpropagation, exhibits spatio-temporal locality. This is much more aligned with learning in humans and animals. Deriving such a framework is a worthwile endeavour, however, the paper could better motivate the implications and applications to modern machine learning. It remains unclear whether the proposed framework could for example lead to new practical and improved learning algorithms for neural networks. Such examples or investigations are missing.  For instance, a small numerical demonstration on a standard learning benchmark such as on-line recognition of MNIST digits could be a way to motivate the approach.

While there may be significant and intriguing findings, it is equally important to present them in a way such that they can be appreciated and understood by the target audience. I encourage the authors to reconsider how to communicate the main results to a broader machine-learning audience. A less technical and more accessible paper might have a greater impact on the field compared to the current one.

After careful consideration and discussion with the reviewers, I recommend to reject the paper and encourage the authors to take the feedback into account to prepare a more accessible variant of the work.

**Justification For Why Not Higher Score:**

In its current form, the paper likely not be understood and appreciated by the wider machine-learning community, and is therefore not ready for publication. It remains unclear why truly local spatiotemporal learning in recurrent neural networks is a strong claim and will have high-impact in machine-learning.

**Justification For Why Not Lower Score:**

N/A

---

### Decision · Program_Chairs · 2024-01-16

Reject